# A Novel DME-YOLO Structure in a High-Frequency Transformer Improves the Accuracy and Speed of Detection

**Zhiqiang Kang \*, Wenqian Jiang, Lile He and Chenrui Zhang**

School of Mechanical and Electrical Engineering, Xi'an University of Architecture and Technology, Xi'an 710055, China; jwt147@163.com (W.J.); hllnh2013@163.com (L.H.); 13225625126@163.com (C.Z.)
\* Correspondence: zqkang@xauat.edu.cn; Tel.: +86-15339232336

**Abstract:** Traditional YOLO models face a dilemma when it comes to dim detection targets: the detection accuracy increases while the speed inevitably reduces, or vice versa. To resolve this issue, we propose a novel DME-YOLO model, which is characterized by the establishment of a backbone based on the YOLOv7 and Dense blocks. Moreover, through the application of feature multiplexing, both the parameters and floating-point computation were decreased; therefore, the defect detection process was accelerated. We also designed a multi-source attention mechanism module called MSAM, which is capable of integrating spatial information from multiple sources. Due to its outstanding quality, the addition of MSAM as the neck of the original YOLOv7 model compensated for the loss of spatial information in the process of forward propagation, thereby improving the detection accuracy of small target defects and simultaneously ensuring real-time detection. Finally, EIOU was adopted as a loss function to bolster the target frame regression process. The results of the experiment indicated detection accuracy and speed values of up to 97.6 mAP and 51.2 FPS, respectively, suggesting the superiority of the model. Compared with the YOLOv7 model, the experimental parameters for the novel DME-YOLO increased by 2.8% for mAP and 15.7 for FPS, respectively. In conclusion, the novel DME-YOLO model had excellent overall performance regarding detection speed and accuracy.

**Keywords:** dim target detection; DME-YOLO model; attention mechanism module; loss function; detection accuracy and speed

## 1. Introduction

A high-frequency transformer is an important magnetic component of electronic circuits. It is widely used as part of a power transformer in computers, displays, smartphones and other common electronic products [1]. Therefore, the quality inspection of a high-frequency transformer is critically important in ensuring product quality control and improving factory efficiency, as well as being an essential part of production. As shown in Figure 1, high-frequency transformer defects can randomly appear on different parts of a device (e.g., folded tape, incomplete label, iron core misalignment, etc.). These defects are ubiquitous in the process of production, posing a potential threat to both production efficiency and the operating costs of enterprises [2–4].

Currently, defect inspections are mostly dependent on artificial visual detection. In practice, the variation in workers' skills and levels of experience inevitably leads to uncertainty in the assessment of production quality. Additionally, manually scrutinizing almost negligible defects is time-consuming and requires a lot of manpower [5]. In contrast with artificial vision inspection, machine vision inspection methods have the advantages of high precision and high efficiency and have been applied in many industrial scenes. Machine vision methods can be categorized into traditional image algorithms and deep learning algorithms [6]. Generally, traditional image algorithms are completely dictated by human-designed features and thus subject to the limited generalization ability in detection. In contrast, in constructing a convolutional neural network, the deep learning method utilizes

a gradient descent algorithm to iteratively learn features adaptively [7]. Therefore, the deep learning method is able to achieve a more robust generalization ability and detection accuracy than its traditional counterpart.

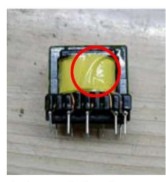 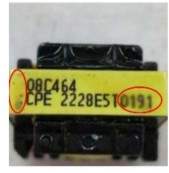 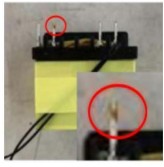 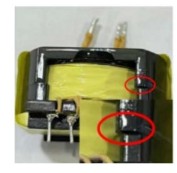 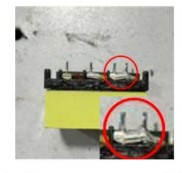 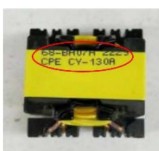

(**a**) Folded tape    (**b**) Blurred label    (**c**) Burned pins    (**d**) Misplaced core    (**e**) Overmany solder    (**f**) Incomplete label

**Figure 1.** Some high-frequency transformers with surface defects.

Redmon, J. et al. first proposed a deep-learning-based defect detection method to meet the demands of current enterprises [8]. Deep learning detection methods [9–11] can be divided into single-stage detection methods and two-stage detection methods [12,13]. The two-stage detection algorithm is represented by the R-CNN series [14,15], which first extracts features based on candidate boxes and then classifies them. Consequently, its detection accuracy is high, while its speed is slow. The single-stage detection method of the target frame is represented by the YOLO [16–18] series algorithm, which integrates the regression of the category and target frame into one forward operation and thus greatly improves the detection speed. However, its detection accuracy for small targets is low [19,20].

The problem lies in the presence of small target defects on the surfaces of products. Due to the small proportion of target defects in the image pixel and the singularity of the feature, the detection process usually has inadequate accuracy. To address the dilemma between accuracy and speed [21], an increasing number of researchers are adding attention mechanisms to their models to improve the detection accuracy of small targets. The principle function of the attention mechanism [22] is to assign different weights to all channels of a certain feature layer and to highlight the channels with effective feature information so that the detection accuracy increases.

In this study, we propose a high-frequency transformer appearance defect detection algorithm named DME-YOLO. DME-YOLO makes features reusable by concatenating Dense blocks into the backbone network with a Dense block structure. Consequently, it considerably reduces the network parameters and floating-point calculations, while ensuring accuracy at the same time. Furthermore, a multi-information source spatial attention mechanism named MSAM is proposed and applied to the neck of the model. Existing attention mechanisms, such as Senet [23], ECA [24], CBAM [25], CA [26], etc., merely focus on the spatial information of the convolution layer, ignoring the coherence of the spatial information within the whole network. By modularizing the process of recording information and applying attention, MSAM is capable of comprehensively considering spatial information from multiple sources when applying attention and compensating for the loss of spatial information in the process of forward propagation, while ensuring the detection accuracy of small target defects. Then, EIOU [27] is treated as the target frame loss function in the decoder part of the model. EIOU not only directly reduces the difference between the width and height of the target frame and the prediction frame, but also improves the convergence speed and regression accuracy of the target frame. Our major contributions are as follows.

(1) A backbone was constructed using an efficient Dense block to expand the receptive field and reduce the number of parameters, thus enabling the network to effectively detect defects on various scales.

(2) A novel attention mechanism called MSAM was designed and applied to the neck of the model. This mechanism is able to extract multiple spatial data points in the neck network to compensate for the small target defect information lost in the process of forward propagation.

(3)  Since EIOU is a superior solution to the problem of aspect ratio loss in target frame regression, we replaced the original CIOU with EIOU as the target frame loss function. This modification improved the convergence speed and stability of the target frame.

(4)  Experiments on appearance defects in a high-frequency transformer dataset verified that DME-YOLO has good performance.

## 2. Related Studies

Industrial detection methods can be divided into traditional machine learning methods and deep learning methods.

### 2.1. Traditional Machine Learning Methods

Before the advent of deep learning methods, machine learning methods were widely used in defect detection tasks [28]. When conducting detection using such methods, Histogram of Oriented Gradients (HOG) [29], Scale-Invariant Feature Transform (SIFT) [30–32] and other methods are commonly utilized to extract features, whereas SVM and Decision Tree [33] are dedicated to the problem of classification. Duan et al. [34] proposed an AdaBoost algorithm with a penalty term to extract features and detect weld defects in pipelines. Suan [35] used Gaussian feature pyramid decomposition to construct a significant feature map to detect strip surface defects. In general, traditional machine learning methods have high detection accuracy when the light source and industrial field environment are stable. However, the algorithm design is complicated and time-consuming, the detection environment is demanding, and there is a lack of generalization ability, leading to the use of deep learning methods as a substitute [36].

### 2.2. Deep Learning Methods

Deep learning methods are superior to traditional machine learning methods with respect to accuracy and generalization ability. The R-CNN series [37] and YOLO series [38] are two of the most representative deep learning algorithms in industrial defect detection.

The R-CNN series has excellent precision [39]. Deqiang He et al. [40] tested the weld quality of the train body and proposed the ABC mask R-CNN algorithm, consisting of phased array ultrasonic detection, which can achieve good detection accuracy. In the work of Duong Huong Nguyen [41], a faster R-CNN algorithm was introduced to detect structural defects in steel plates. The results revealed that the method was robust in terms of positioning accuracy. For 80% of the test scenarios, the overlap rate between the predicted boundary box and the actual damaged area was over 40%. However, it is too computationally taxing to successively extract features using the R-CNN algorithm with a large number of candidate boxes, and thus the detection speed is slow.

The YOLO series algorithm performs effectively in terms of real-time industrial inspection [42]. This algorithm incorporates target location and category prediction into one regression task, which considerably reduces the time loss of feature extraction. Based on YOLOv5s, Yi Li et al. [43] designed a real-time detection system for packaging defects, with a detection speed of 30 FPS. To lessen the complexity of the network, Xue et al. [44] introduced a coal gangue detection algorithm, with YOLOv3 as its basic model and ResNet18 as its backbone network. Additionally, unstructured pruning technology served as a means to reduce the amount of computation in the model. As a result, this model reached 96.27% accuracy and 22 FPS in the measurement of speed [45]. Ma et al. [46] put forward a lightweight detector based on YOLOv4 for surface defect detection in aluminum strips. By applying dual channel attention, the whole network was able to take more detailed information into account, while simultaneously improving the detection accuracy. Li et al. [47] adopted YOLOv5 for the defect detection of insulators in a power grid and adopted light correction technology for image preprocessing to improve the model's ability to detect defects. In summary, the YOLO algorithm has strong real-time performance. However, it is somewhat lacking in accuracy, and solving this could require compensation via an attention mechanism, image preprocessing and other technologies. Inspired by this, we

propose an MSAM-YOLO algorithm to further improve the accuracy in real-time industrial inspection scenarios.

## 3. Methods

### *3.1. YOLOV7 Object Detection*

#### 3.1.1. YOLO's Workflow

YOLO is a single-stage object detection algorithm with a good detection speed and accuracy. In the process of detection, the image is first reshaped to a multiple of 32, usually 416 × 416 or 640 × 640, within reason and considering the subsequent calculation. Then, the grid is divided over the image into three scales: 13 × 13, 26 × 26 and 52 × 52. This refinement of the grid could render small targets easier to detect. In each grid, the K-means clustering algorithm is applied to obtain three prior boxes, the probability (confidence) of the existence of the target in the grid and the category of the target. With repeated iterations, candidate boxes are obtained after the confidence and category are accurately fitted to the true value, and the width, height and center point are fully adjusted in the prior box. Ultimately, according to the confidence, non-maximum suppression (NMS) is conducted on all candidate boxes, and the final prediction results are output.

#### 3.1.2. Network Architecture of YOLOv7

As shown in Figure 2, YOLOv7 [48] is a fully convolutional network with all convolutionally computational modules. The network architecture is composed of three parts, including the feature extraction part (backbone), the feature fusion part (neck) and the decoder (head).

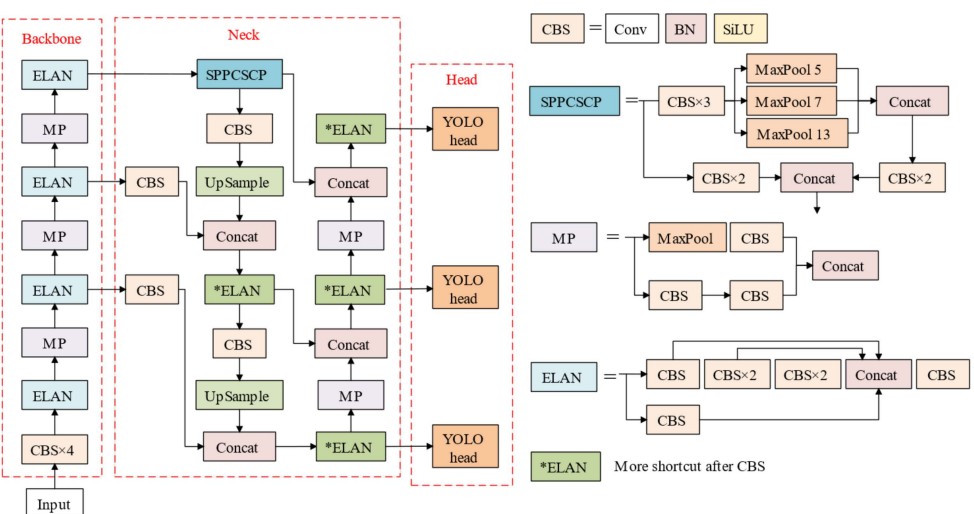

**Figure 2.** The original architecture of YOLOv7.

A backbone is used to extract the feature information from images and provides the basis for the prediction of the head. CBS is a collection of convolution (Conv), batch normalization (BN), and Silu activation functions. MP downsamples the two routes using maximum pooling and convolution (stride = 2), respectively, and then summarizes the information via concat operations. The model for these processes retains the most effective features. ELAN consists of multiple CBS modules connected by multiple residuals to prevent the network from degradation and deepen the network layers to extract deep features. The extracted features are then fed into the neck for fusion. The neck part contains three scales, which transmit information between each other via the upsampling process and MP (subsample structure), so that the data points on different scales can be fused with each other to enrich the features of the model. The head part consists of the REP module and the CBM module, makes the final adjustment to all information and outputs it in the form of the feature layer.

### 3.2. The Proposed DME-YOLO

In order to improve the real-time precision of the online detection of small defects, the DME-YOLO method is proposed, and the structure is shown in Figure 3. In the backbone, the Dense block is used for feature extraction to reduce the model size and floating-point computation [49–52]. In addition, a Multi-Source Space Attention Module (MSAM) is added to the neck, enabling the model to pay more attention to the changes in the spatial information of the neck and adjust the spatial weight of the feature layer. Consequently, the network becomes more effective in classifying and locating defects. For headers, we define the target frame loss function for DME-YOLO.

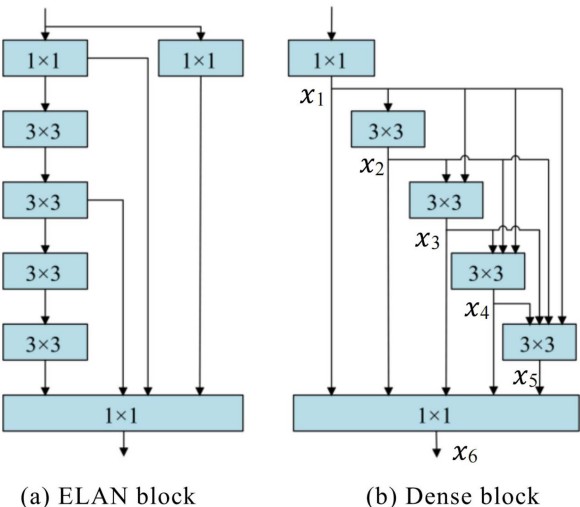

(a) ELAN block        (b) Dense block

**Figure 3.** Comparison between the ELAN block and Dense block.

### 3.2.1. Building a New Backbone

A backbone is responsible for image feature extraction and is thus critical in recording detection results. The backbone of the original model extracted deep features through a series of repeated stacking of convolutions, which significantly increased the number of parameters and the computational costs of the backbone. To tackle these problems, a Dense block is applied to build a new backbone. As demonstrated in Figure 3, the input is first reshaped by a $1 \times 1$ convolution in the Dense block, and it then passes through multiple $3 \times 3$ convolution. With xi representing the output of each layer of convolution and $H_i$ denoting the non-linear operation process, Formula (1) is defined as follows:

$$x_i = H_i \left([x_1, x_2, \ldots, x_{(i-1)}]\right) \tag{1}$$

The input of each convolutional layer of the Dense block is the sum of the outputs from all previous convolutional layers, with a residual connection existing in each convolutional layer. However, the problem is that the residual connection of ELAN blocks only exists in the last convolution layer. We found that a Dense block could be interpolated from the perspective of the features. Following this thread, building dense convolution modules through dense connections between convolutional layers in the process of network downward transmission could become a more reusable option. Therefore, in contrast with ELAN, a Dense block was found to be more efficient in terms of feature reuse, and it considerably reduced the number of parameters and computation costs.

### 3.2.2. Simplifying the Original Neck Structure

The neck is responsible for processing the feature map from the backbone. MSAM was proposed as an additional structure in the neck of the network in order to improve the detection effect. The attention mechanism could adjust the channel or spatial weight of the feature map to highlight the effective features. Common attention mechanisms (e.g., SE and CBAM, etc.)

only focus on the current feature map while ignoring the changes in feature information in the process of network transmission. The shortcomings of traditional mechanisms inspired us to design MSAM as a multi-source structure. As shown in Figure 4, there are two basic modules in MSAM: the recording module and the attention module.

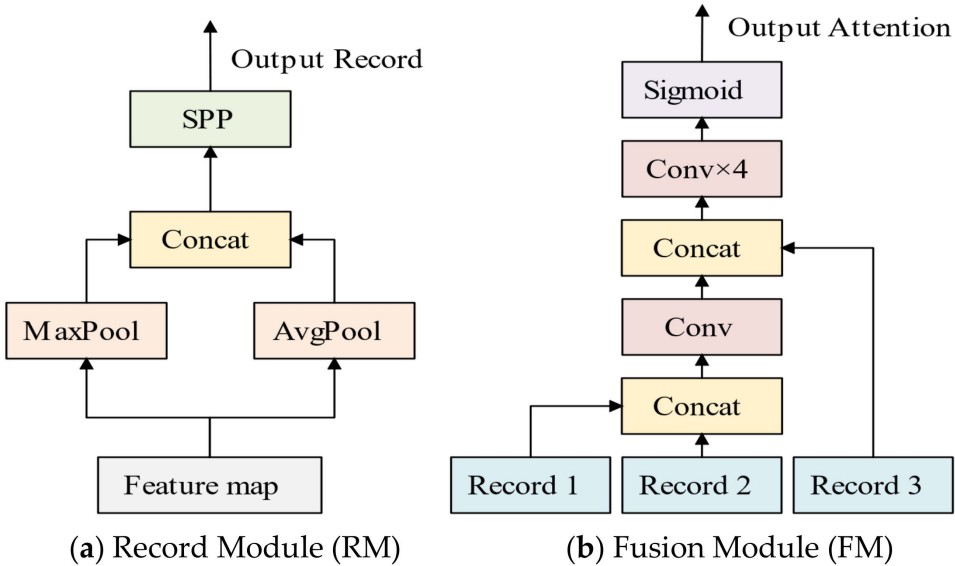

**Figure 4.** Multi-Source Space Attention Module.

The recording module was designed to extract the spatial information from any feature map. First, we maximized and averaged the feature maps using all channels. Afterward, we spliced the channel direction and applied the SPP structure [53], which comprised pooling layers of sizes 5, 9, 13 and thus could be used to expand the channels in the feature map by a factor of 4. The application of the SPP structure increased the number of channels, while also expanding the receptive field. Finally, we obtained record information as the output. The fusion module could integrate information from three recording modules. First, record 1 and record 2 were joined in the channel dimension through the convolution process (kernel = 7, stride = 2), and the channel was halved to complete the first integration of spatial information [54]. Then, the result of the first integration was concatenated with record 3 and convolved four times (K = 7, S = 2) so that the number of channels was reduced to one. Finally, Sigmoid activation enhanced the nonlinear fitting ability and output an attention weight.

As shown in Figure 5, MSAM was applied to the neck, which included four FM modules and eight RM modules. For simplicity, we streamlined the original Figure 5 neck structure. Eight RM modules were distributed before and after the processes of upsampling and downsampling Figure 6, where the feature information significantly changed in order to record the spatial information of the neck. The four FM modules were located at the back to apply spatial attention to the neck. The FM received three messages from the RM: the current feature graph of the FM (3), the previous text (1) and the context (2).

The application of the FM not only prevented the loss of spatial information in the process of downward transmission in the network, but also highlighted effective features and improved the overall detection accuracy.

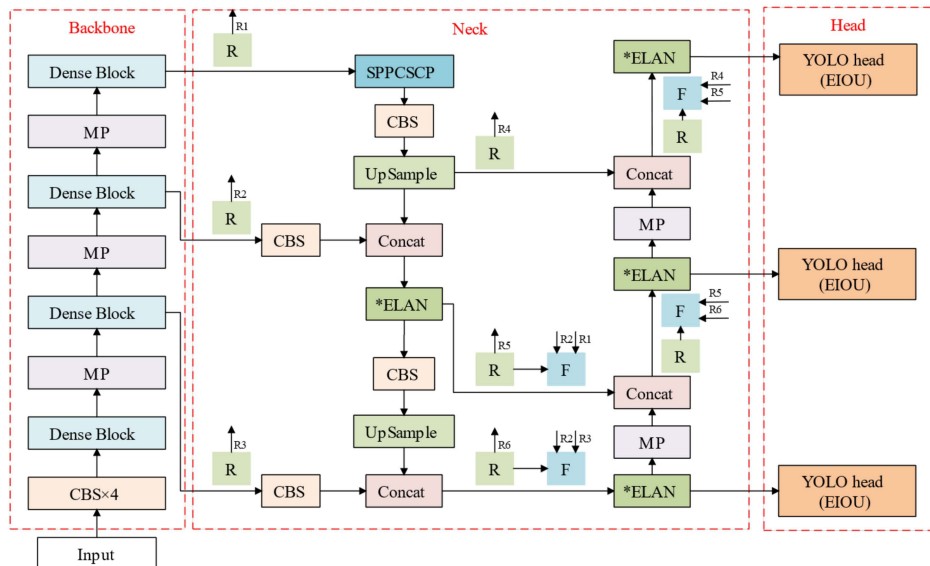

**Figure 5.** The MSAM-YOLO architecture.

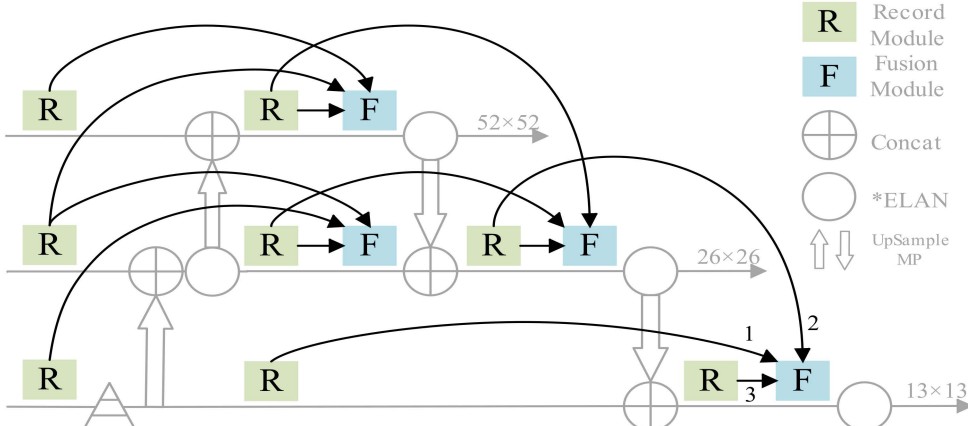

**Figure 6.** The overall architecture of MSAM.

### 3.2.3. Head Optimization

The head is the feature map to be decoded, which entails defect-type information in the prediction box and confidence levels. The deviation between the real box and the predicted box is defined by the loss function. In order to thoroughly describe the target frame loss, some researchers have introduced a function based on *IOU* loss (e.g., aspect ratio loss). As shown in Equation (2), the loss was calculated by using *EIOU*, which consists of the *IOU* loss, center distance loss and width and height loss. Compared with *CIOU*, the width and height loss of *EIOU* can minimize the difference between the width and height of the target frame and the forecast frame better than its *CIOU* counterpart. Therefore, the minimization of the divergence between the target frame and forecast frame accelerates the convergence process, which makes the convergence speed faster. The following Equations (3)–(5) show the formulae for *EIOU* loss.

$$Loss_{EIOU} = L_{IOU} + L_{dis} + L_{asp} \tag{2}$$

$$L_{IOU} = 1 - IOU \tag{3}$$

$$L_{dis} = \frac{\rho^2\left(b, b^{gt}\right)}{\left(w^c\right)^2 + \left(h^c\right)^2} \tag{4}$$

$$L_{asp} = \frac{\rho^2\left(w, w^{gt}\right)}{\left(w^c\right)^2} + \frac{\rho^2\left(h, h^{gt}\right)}{\left(h^c\right)^2} \tag{5}$$

where $b^{gt}$ and $b$ represent the central coordinates of the prediction box and the real box, respectively. $\rho^2\left(b, b^{gt}\right)$ denotes the distance between the center point of the predicted frame and the real frame. $w$ and $h$ are the width and height of the real box, while $b^{gt}$ and $h^{gt}$ are the width and height of the prediction box, respectively. $w^c$ and $h^c$ are the width and height of the smallest outer box that includes the real box and the predicted box.

## 4. Experiments and Results

### 4.1. High-Frequency Transformer Defect Datasets

To verify the performance of DME-YOL, the datasets were manually annotated via multiple annotators using the lableImg software (gitee.com), as shown in Table 1. According to factory production standards, we adopted the high-frequency transformer appearance defect dataset, including six categories of defects required for detection. The terms of these defects are as follows: iron-core rejection (ir), pin rejection (pr), solder rejection (sr), tape rejection (tr), label rejection (lr) and label acceptance (la). To ensure balanced categories, 270~310 images of each defect were selected, each 640 × 640 in size. We increased the original dataset to 5316 images after horizontal and vertical flipping operations (mark 6972). As is conventional, we separated these images into the training set and test set in a ratio of 9 to 1. Hence, 4784 images were used for training and 532 images were used for testing. A K-means algorithm was used to cluster the annotation boxes of the dataset, and we obtained nine prior boxes. Their coordinates were (12, 16), (19, 36), (40, 28), (36, 75), (76, 55), (72, 146), (142, 110), (192, 243), (459, 401).

**Table 1.** Categories in high-frequency transformer dataset.

| Category | Annotation Standard | Number of Boxes | Examples |
|----------|--------------------|-----------------|----------|
| skr | Damage to the skeleton or misalignment of the skeleton assembly. | 305 |  |
| pr | The pin is scalded by high temperatures and turns golden brown, or the pin is stuck to the solder. | 278 |  |
| sor | Too little solder leads to exposed wiring and too much solder leads to tinning. | 282 |  |
| tr | The adhesive cloth is wrinkled, warped, scalded due to high temperatures, stained or damaged. | 290 |  |

**Table 1.** *Cont.*

| Category | Annotation Standard | Number of Boxes | Examples |
|---|---|---|---|
| lr | The label is incomplete; the font is distorted or stained. | 299 | |
| la | Centered label, complete, parallel, free of stains and breakage. | 289 | |

### 4.2. Experimental Setup

Pytorch was applied as the framework to examine the model in the training and testing process. The experimental environment was configured as follows: Intel i7 6700 CPU, NVIDIA RTX3080ti 12 G GPU and Windows 10 operating system. The size of the input image was 416 × 416. To optimize our model and obtain better performance, we employed the Adam optimizer along with the cosine annealing learning strategy, instead of sgd, since the second-order momentum in the Adam optimizer converged faster than sgd. The initial learning rate was set to 0.001, with 0.00001 as its lower bound. Figure 7 shows that Mosaic data augmentation was adopted to splice four images by means of random scaling, random cropping and random arrangement. After its augmentation, the detection dataset was supplemented. However, the complementation of randomly generated new small targets remarkably reduced the capacity of the model to detect small targets.

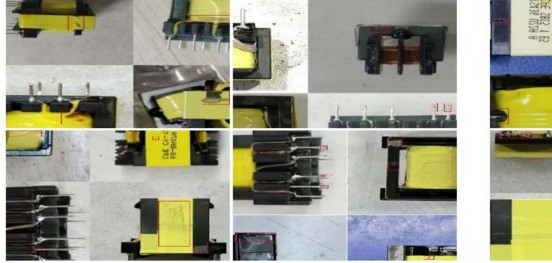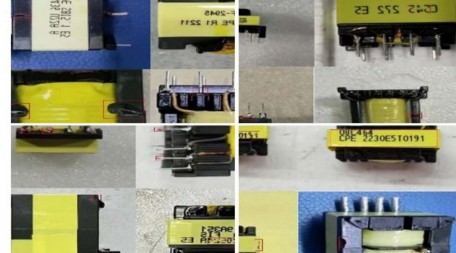

**Figure 7.** Mosaic data augmentation.

### 4.3. Performance Evaluation

Regarding defect detection, it is irrefutable that the importance of the training speed is comparable to that of precision. To measure the precision of the model, we adopted the mean average precision (*mAP*) as a metric. The calculation of *mAP* involved correlating it with the accuracy and recall rate. Then, 0.5 was set as the threshold of the IOU, meaning that the prediction box should be recognized as positive only if the intersection ratio between the true box and the prediction box was greater than 0.5; otherwise, it was judged as negative. The *mAP* was calculated according to Equations (6)–(8):

$$Precision = \frac{TP}{TP + FP} \tag{6}$$

$$Recall = \frac{TP}{TP + FN} \tag{7}$$

$$mAP = \frac{\sum_{i=1}^{c} \int_0^1 P(R)dR}{c} \qquad (8)$$

where *TP* is the number of correctly detected defect samples; *FP* is the number of detected non-defective samples; *FN* is the number of defect samples in which errors are detected; and *P* and *R* represent accuracy and recall, respectively. In addition to the *mAP* metric for precision evaluation, frames per second (FPS) was used as a speed indicator to evaluate the real-time performance of the model.

### 4.4. Comparisons with Other Methods

DME-YOLO was used to detect the existing defects in the dataset of high-frequency transformer images. The results of verification involved target frame location, type information and confidence. As Figure 8 shows, the model had extraordinary holistic detection performance for small target defects (e.g., pin rejection and core rejection). To evaluate the effectiveness of the model, two lightweight models (YOLOv4-tiny and YOLOv5s), two deep convolution models (YOLOv7 and Faster-RCNN) and three models for industrial defect detection [52–54] were reproduced and verified.

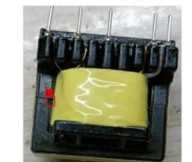 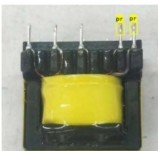 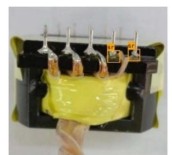 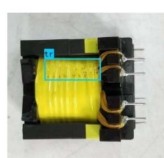 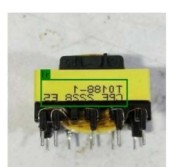 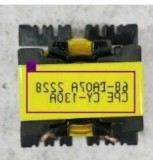

(**a**) iron-core rejection   (**b**) pin rejection   (**c**) solder rejection   (**d**) tape rejection   (**e**) label rejection   (**f**) label acceptance

**Figure 8.** Results of detection.

The data shown in Table 2 suggest that the proposed model achieved the highest mAP (2.8% higher than YOLOv7) and an acceptable improvement in FPS (15.7 higher than YOLOv7). In addition to the comparison with the baseline YOLOv7, the mAP measurement of DME-YOLO was 97.6% (6.7% better than that of YOLOv5s and 5.4% better than that of YOLOv4-tiny). Furthermore, DME-YOLO had 3.4% greater mAP and was 2.75 times faster in terms of FPS than Faster-RCNN. Unlike the DCC-CenterNet, DEA_RetinaNet and RDD-YOLO methods, the performance of the model in the detection of small target defects such as ir, pr and sr was surprisingly excellent. This was mainly due to the Mosaic method, the MSAM attention mechanism and the use of EIOU.

**Table 2.** Detection results of state-of-the-art methods.

| Methods | mAP | Ir | pr | sr | tr | lr | la | FPS |
|---|---|---|---|---|---|---|---|---|
| YOLOv5s | 90.9 | 86.3 | 83.2 | 84.0 | 97.0 | 95.1 | 99.8 | 62.4 |
| YOLOv4-tiny | 92.2 | 88.5 | 84.0 | 87.3 | 97.1 | 96.6 | 99.8 | 89.2 |
| Faster-RCNN | 94.2 | 88.4 | 93.1 | 88.5 | 99.6 | 95.5 | 99.9 | 18.6 |
| YOLOv7 | 94.8 | 91.1 | 90.8 | 88.1 | 99.8 | 98.9 | 100.0 | 35.5 |
| DCC-CenterNet [52] | 95.9 | 94.0 | 92.7 | 91.5 | 99.8 | 97.7 | 99.9 | 68.4 |
| DEA_RetinaNet [53] | 96.0 | 92.8 | 95.4 | 89.2 | 99.9 | 98.9 | 100.0 | 15.2 |
| RDD-YOLO [54] | 96.4 | 94.8 | 93.6 | 91.6 | 99.8 | 98.5 | 100.0 | 48.1 |
| DME-YOLO | 97.6 | 95.1 | 98.1 | 94.5 | 99.8 | 97.9 | 99.9 | 51.2 |

### 4.5. Grad-CAM Analysis Method

Gradient-Weighted Class Activation Mapping (Grad-CAM) is a practical visualization method. The fundamental idea of Grad-CAM is that the weight of each channel can be calculated by globally averaging the gradient of the last convolutional layer, given that the output feature map of the last convolutional layer has the most influence on the classification result. These weights are then used to weight the feature map. Finally, a Class Activation Map (CAM) is generated, where each pixel represents the importance of the

pixel region in the classification result. As Figure 9 depicts, the Grad-CAM method was adopted to analyze the detection results of YOLOv7 and DME-YOLO.

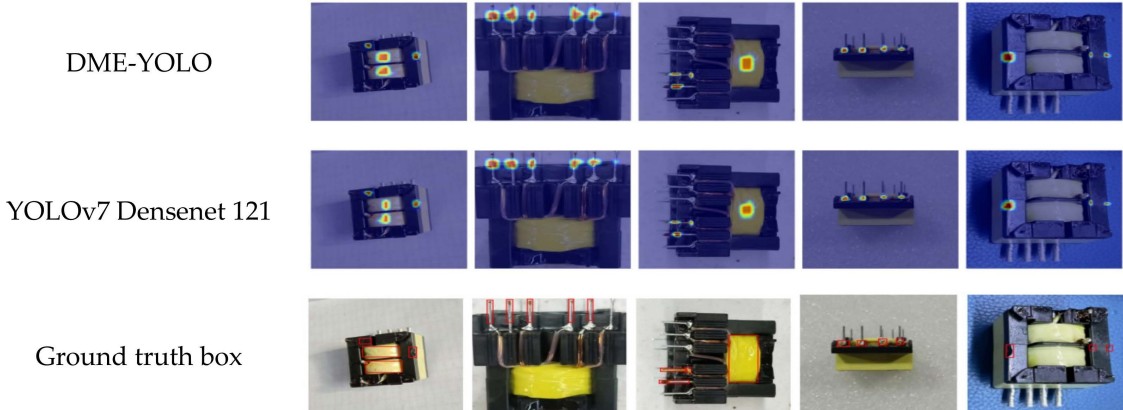

**Figure 9.** Grad-CAM thermal region.

The model was more robust than YOLOv7. The thermal zones of adhesive tape and seal expanded, while those of two types of small targets, such as solder and core, shrunk. In short, the generated thermal regions were closer to the real labeled region, in which the accuracy of the location frame and type of confidence in the prediction frame were significantly enhanced.

*4.6. Ablation Study*

4.6.1. Experiments on DME-YOLO

In this stage, an ablation experiment on the improved backbone, neck and head was conducted to verify the effectiveness of the improvement. The performance of DME-YOLO along with its parallel models is listed in Table 3. In this context, D-YOLO, DM-YOLO, DE-YOLO and DME-YOLO denote YOLOv7 with an extra Dense block; YOLOv7 with a Dense block and MSAM; YOLOv7 with a Dense block and EIOU; and YOULOv7 with a Dense block, MSAM and EIOU, respectively.

**Table 3.** The ablation experiments on DME-YOLO.

| Methods | mAP | ir | Pr | sr | tr | lr | la | FPS |
|---------|-----|-----|-----|-----|-----|-----|-----|-----|
| YOLOv7 | 94.8 | 91.1 | 90.8 | 88.1 | 99.8 | 98.9 | 100.0 | 35.5 |
| D-YOLO | 94.9 | 91.2 | 90.5 | 89.2 | 99.9 | 98.6 | 99.9 | 52.3 |
| DM-YOLO | 95.7 | 94.6 | 91.8 | 89.1 | 99.8 | 98.9 | 100.0 | 51.6 |
| DE-YOLO | 97.1 | 93.7 | 96.4 | 94.9 | 99.9 | 97.9 | 99.9 | 52.5 |
| DME-YOLO | 97.6 | 95.1 | 98.1 | 94.5 | 99.8 | 97.6 | 99.9 | 51.2 |

4.6.2. Experiments on MSAM

In this section, we describe an ablation experiment, which was conducted to gauge the impact of the number of MSAM modules on the overall accuracy. MSAM was characterized by its modular and multi-source quality, which enabled us to integrate it extensively into our networks. To further explore this quality and its impact on the model, it was hypothesized that there was a correlation between the quantity and accuracy. Therefore, confirming the existence of the correlation was critical in validating the viability of the model. As shown in Figure 10, four locations on the neck were selected to add attention. We successively increased the number of MSAM from P1 to P4 in each experiment, and the results indicated that the addition of MSAM could increase the accuracy.

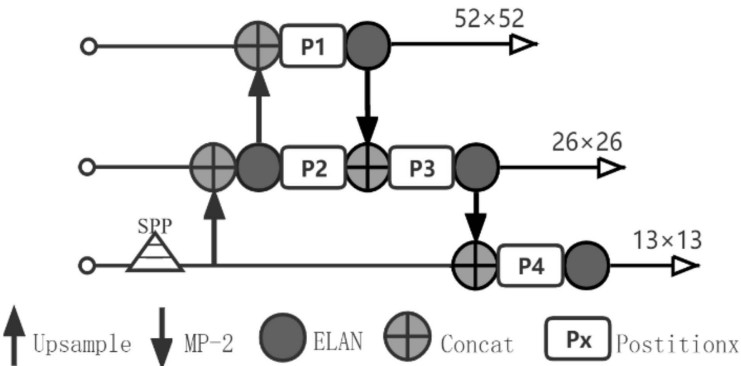

**Figure 10.** Four locations on the neck for attention.

The experimental results (in Table 4) indicate that the mAP declined after adding the second and third SAM modules, which could have been caused by a lack of coordination between the various SAMs. With reference to Figure 10, the influence of the attention module on the output of the neck could be analyzed. SAM at P1 was assumed to exert a positive influence on the convolution layer and transmit to P3 with downsampling, while SAM at P3 was only concerned with its own spatial information at this time. Therefore, it simultaneously exerted an influence on the convolution layer to counteract the positive influence transmitted at P1 and even became negative. However, MSAM showed the opposite effect. MSAM at P3 introduced spatial information from P1 and P2 and added its own spatial information, and thus coordination considerations could be made when adjusting the spatial weights. A similar conclusion could be drawn from the data in Table 4, namely that the accuracy of the network improved with the increase in the number of MSAM. After the introduction of three SAMPlus modules, the accuracy of SAM was fully exceeded. MSAM's multi-information source scheme played a role in promoting the flow of information in cyberspace and thereby enabled it to focus on the changes in spatial information in a wider local context. Therefore, the use of multiple MSAMs could further expand this range so that it covers the entire neck.

**Table 4.** Experiments on MSAM and SAM.

| Module | P1 | P1~P2 | P1~P3 | P1~P4 |
| --- | --- | --- | --- | --- |
| MSAM | 95.1 | 95.4 | 96.8 | 97.6 |
| SAM | 95.6 | 95.5 | 92.5 | 93.8 |

4.6.3. Experiments on Loss Functions

In the experiments, a variety of *IOU* loss functions were chosen, including *GIOU* loss, *CIOU* loss, *EIOU* loss and *SIOU* loss. The loss function that was the most favorable in improving the performance of the model was also a topic of discussion. *GIOU* loss intro-duces a minimum outsourcing frame to quantify the distance between two frames, thus solving the problem of zero gradient in the case of no intersection between two frames. Based on *GIOU*, *CIOU* considers the overlap degree and size of the two boxes, rendering the regression process more stable. Grounded in *CIOU*, *EIOU* refines the aspect ratio loss and accelerates the process of target frame regression. Additionally, *SIOU* not only considers the direction of regression, but also introduces the angle parameter to make the prediction frame return to the axis nearest to the real frame, thus effectively reducing the freedom of the target frame regression. The formulas for *GIOU* loss, *CIOU* loss and *EIOU* are shown in Equations (9)–(11), respectively.

$$L_{GIOU} = 1 - IOU + \frac{\rho^2\left(b, b^{gt}\right)}{c^2} \tag{9}$$

$$L_{CIOU} = 1 - IOU + \frac{\rho^2(b, b^{gt})}{c^2} + \alpha\beta \tag{10}$$

$$L_{EIOU} = 1 - IOU + \frac{\rho^2(b, b^{gt})}{c^2} + \frac{\rho^2(w, w^{gt})}{(w^c)^2} + \frac{\rho^2(h, h^{gt})}{(h^c)^2} \tag{11}$$

where $\beta = \frac{4}{\pi^2}\left(arctan\frac{w^{gt}}{h^{gt}} - arctan\frac{w}{h}\right)$ is defined to measure the aspect ratio loss. $\alpha$ is the weight parameter. $b^{gt}$ indicates the center position coordinates of the prediction box. $b$ represents the center point coordinates of the real box. $w$ and $h$ are the width and height of the real box, respectively. $w^{gt}$ and $h^{gt}$ are the width and height of the prediction box. $w^c$ and $h^c$ denote the width and height of the smallest outer box, which includes the real box and the predicted box, respectively.

The experimental results are shown in Table 5. *EIOU* was chosen as the target frame loss function of the model to ensure the stability and effectiveness of the regression process, since it comprehensively considered the *IOU* loss, center distance loss and aspect ratio loss, thereby significantly improving the accuracy.

**Table 5.** Comparison of different loss functions.

| Loss Function | *GIOU* | *CIOU* | *EIOU* | *SIOU* |
|---|---|---|---|---|
| mAP | 83.8 | 83.6 | 84.0 | 82.4 |

## 5. Conclusions

In this study, a new model named DME-YOLO was designed for the detection of defects in the appearance of a high-frequency transformer. The newly designed model was based on YOLOv7, and several modifications and improvements were focused on the trunk, neck and head of the model. The experiments showed that the model had good performance in terms of both detection accuracy and speed. More specifically, in order to enhance the ability of feature extraction, the acceptance domain and adopted Dense block were expanded as the main components of the DME-YOLO backbone. In the neck, MSAM was introduced in the form of two modules, and spatial information from multiple sources was fused to obtain rich semantic information. In the head, EIOU was employed to improve the accuracy of the target box regression. To verify its robustness and generalization, a high-frequency transformer appearance defect dataset was used for the experiments. The final results demonstrate that DME-YOLO achieved 41.2 FPS and 97.6 mAP, remarkably outperforming other state-of-the-art models in the same experimental scenario. However, there remains room for the improvement of RDD-YOLO regarding some small defects (e.g., water and oil stains on the tape). In future research, image preprocessing strategies or a stronger backbone should be adopted to further improve the network. In addition, a model with pruning algorithms could be developed that is lightweight and maintains high accuracy.

**Author Contributions:** Z.K.: Conceptualization, Methodology, Writing—Reviewing and Editing. W.J.: Data Curation, Writing—Original Draft Preparation, Visualization. L.H.: Supervision. C.Z.: Software, Validation. All authors have read and agreed to the published version of the manuscript.

**Funding:** This work is supported by the Natural Science Foundation of Shaanxi Province (Grant No. 2019JM-286) and Xi'an Municipal Bureau of Science and Technology (Grant No. 2021JH-05-0071).

**Data Availability Statement:** The datasets analyzed during the current study are available in the Dataverse repository CV Datasets on the web (cvpapers.com). http://www.gepperth.net/alexander/interests.html#carbenchmark, http://mmlab.ie.cuhk.edu.hk/projects/TCDCN.html, https://boxy-dataset.com/boxy/ and http://www-sop.inria.fr/members/Alexis.Joly/BelgaLogos/BelgaLogos.html. Data regarding the appearance defects of a high-frequency transformer are available only to collaborating scientists from the respective participating centers. The data may be available upon request to some of the participating centers but not for all due to relevant data protection laws. Data sharing is not applicable to this article as no datasets were generated or analyzed during the current study. All authors confirm that the data supporting the findings of this study are available within the article.

**Conflicts of Interest:** The authors declare no conflict of interest.

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
