# Peer review of "A Novel DME-YOLO Structure in a High-Frequency Transformer Improves the Accuracy and Speed of Detection"

_electronics, doi:10.3390/electronics12183982_

Round 1

Reviewer 1 Report

The article is devoted to the modification of the YOLov7 algorithm for the detection of dim targets. The topic of the article is relevant, since improving the accuracy in detecting dim targets is an important task that has a wide practical application, for example, in computer vision systems in electronics. The authors give a sufficient overview, the results are clearly described, however, there are some comments that would improve this article.

1. Modification of the authors' algorithm allows us to improve the accuracy of detecting dim objects. This is also interesting from a fundamental point of view. However, the authors use the YOLov7 algorithm, now a new YOLov8 algorithm has already been released, which works more accurately. A comparison of the modified algorithm with the stacked YOLov8 may also be of interest.

2. Electrical components were used as an object of study for recognition. Can the modified method claim to be universal and be used to recognize other classes of dim objects?

In general, the article is of scientific and practical interest and can be published after a little revision.

Author Response

  1. Modification of the authors' algorithm allows us to improve the accuracy of detecting dim objects. This is also interesting from a fundamental point of view. However, the authors use the YOLov7 algorithm, now a new YOLov8 algorithm has already been released, which works more accurately. A comparison of the modified algorithm with the stacked YOLov8 may also be of interest.

Answer: We are currently undergoing a lot of testing and verification, because the verification time is relatively long, we believe that YOLOv8 will be better.

  1. Electrical components were used as an object of study for recognition. Can the modified method claim to be universal and be used to recognize other classes of dim objects?

 Answer: The modified method is universal and be used to recognize other classes of dim objects, the recognition of electronic devices is only an application object.

Reviewer 2 Report

The manuscript titled "A novel DME-YOLO structure for high frequency transformer improves the accuracy and speed of detection" introduces the DME-YOLO deep learning model designed explicitly for identifying defects in high-frequency transformers. The integration of a DenseNet backbone, a multi-source attention module, and an enhanced loss function synergistically heightens both the accuracy and speed of defect detection. Empirical tests underscore its good performance.

However, I have the following major concerns,

(1)To begin with, the manuscript would benefit significantly from thorough proofreading, as the current level of grammatical and typographical errors detracts from a comprehensive understanding of the content. Such oversights do raise concerns about the diligence with which the research was conducted. As an illustration, even the title has typographical errors, missing the "f" in "frequency" and lacking “T” for "Transformer". While I'll highlight specific errors in the subsequent sections, I strongly recommend a rigorous review to enhance the clarity and credibility of the paper.

(2) Further, I would like to address potential discrepancies regarding data availability. Between lines 398 to 400, the authors state, “Data availability: The datasets analyzed during the current study are available in the Dataverse repository: https://www.cs.toronto.edu/~kriz/cifar.html. These datasets were derived from the following public domain resources: http://www.image-net.org/, https://aws.amazon.com/cn/datasets/.” Given my familiarity with commonly used datasets in AI, I've observed that the CIFAR dataset, referenced in the aforementioned URL, does not appear to be utilized in this study. I would urge the authors to revisit this section for accuracy and ensure that the datasets mentioned align with those actually used in the paper. 

(3) Regarding the dataset annotations, clarity is essential. My experience suggests that when working with object detection frameworks like YOLO, human annotation is crucial for generating ground truth. Were multiple annotators involved in this process? And if so, was there an established criterion or standard to ensure consistency in labeling the defects? Since YOLO operates on the principle of supervised learning, ensuring that data is Independent and Identically Distributed (i.i.d.) is of paramount importance. The absence of this detail in the manuscript raises questions about the reliability of the presented results. I strongly recommend the authors to address this critical aspect by adding more details about the labeling process to enhance the credibility of the study.

(4) Regarding the Literature review section, I observed that the content might benefit from more recent updates. While one-stage and two-stage object detection frameworks were indeed considered state-of-the-art in 2020, given that we're now in 2023, it's pivotal to consider recent advances in the domain. In particular, Transformer-based object detection and segmentation models have gained significant attention. For instance, the Segment Anything Model (SAM) from Meta AI presents a compelling approach to object detection. I would highly recommend the authors to update their literature review section to incorporate these recent developments. Additionally, considering a comparative evaluation with the Segment Anything Model (SAM) might elevate the novelty of your work, potentially capturing the interest of a broader readership.

(5) In reference to the multiple-source space attention module depicted in Figure 5, I'm curious about the rationale behind the chosen architecture. Is there a specific motivation or empirical evidence suggesting its superiority for the task at hand? Given the advancements in Neural Architecture Search (NAS), it's conceivable that more optimal architectures might be attainable. It would be enlightening to see additional empirical evidence or discussions that underline the effectiveness of this particular structure in achieving the paper's objectives.

(6) In lines 285 to 287, the manuscript states, “Set the IOU threshold to 0.5, which means that a prediction box is considered a correct prediction only if the intersection ratio between the true box and the prediction box is greater than 0.5.” While this is a standard approach, I'd like to delve deeper into a scenario: Suppose there's a single ground truth box, and the model predicts four boxes. Assuming non-max suppression doesn't reduce them and each of these predicted boxes overlaps with the ground truth box with an IOU > 0.5, would they all be deemed correct? In conventional computer vision practices, this might be accepted since the emphasis is on prediction quality. However, in fields like electronics where physical interpretations are crucial, such outcomes might not be ideal. For instance, while one predicted box correctly matches the target, the other three could be perceived as noise, offering no additional informational value. It would be beneficial if the authors clarify their stance and specify whether they're adhering to the norms of the computer vision community in this regard.

(7) In the manuscript, I did not identify any specific mention or citation regarding the YOLO v7 framework upon which your model appears to be constructed. I strongly encourage the authors to duly acknowledge the source of the code they adapted to develop their model. If the model was developed from scratch, it would be surprising. Training a model from scratch, especially on comprehensive datasets like COCO, is both time-consuming and resource-intensive. Proper citation and acknowledgment of prior work are crucial for scholarly integrity.

Below are some highlighted grammatical errors besides the title error mentioned above; however, this isn't an exhaustive list. I strongly recommend a thorough proofreading of the document.

(1)In the abstract, lines 9 to 13, please avoid capitalizing words after a comma. Additionally, the sentences in this section require restructuring for clarity and coherence.

(2)While it's a minor point, I noticed some authors have used email addresses from domains like 163.com and qq.com. Considering all authors are affiliated with Xi’an University of Architecture and Technology, is there a specific reason for not using educational email addresses?

(3) In line 31, it appears that the Chinese character “、” has been used. Please replace it with the standard English comma for consistency.

(4) In line 115, the sentence "Duong Huong Nguyen[30] The Faster R-CNN algorithm is used to detect structural defects of steel plates." should be revised for clarity. Perhaps it could read: "Duong Huong Nguyen [30] employed the Faster R-CNN algorithm to detect structural defects in steel plates."

(5) In line 161, the phrase "Can retain the most effective features." should be revised for completeness. It might read: "The proposed model can retain the most effective features."

(6) In lines 257 to 253, the mathematical notations "? 2 (?, ? ??)", "w^gt", and "h^gt" appear to be incomplete or incorrectly formatted. Please ensure that these notations are properly presented and defined in the text.

I strongly recommend a thorough proofreading of the document.

Author Response

(1)To begin with, the manuscript would benefit significantly from thorough proofreading, as the current level of grammatical and typographical errors detracts from a comprehensive understanding of the content. Such oversights do raise concerns about the diligence with which the research was conducted. As an illustration, even the title has typographical errors, missing the "f" in "frequency" and lacking “T” for "Transformer". While I'll highlight specific errors in the subsequent sections, I strongly recommend a rigorous review to enhance the clarity and credibility of the paper.

Answer: I sincerely apologize for the problems in the composition of the paper. In the future, I will be strict with myself and proofread the manuscript carefully.I have carefully corrected all the issues you pointed out.

(2)Further, I would like to address potential discrepancies regarding data availability. Between lines 398 to 400, the authors state, “Data availability: The datasets analyzed during the current study are available in the Dataverse repository: https://www.cs.toronto.edu/~kriz/cifar.html. These datasets were derived from the following public domain resources: http://www.image-net.org/, https://aws.amazon.com/cn/datasets/.” Given my familiarity with commonly used datasets in AI, I've observed that the CIFAR dataset, referenced in the aforementioned URL, does not appear to be utilized in this study. I would urge the authors to revisit this section for accuracy and ensure that the datasets mentioned align with those actually used in the paper.

  • Answer:Thank you very much for the reviewer's strict attitude and profound knowledge. The questions you raised in this paper are very worthy of my study and research, and I would like to express my special thanks!  Some datasets have been applied during the research process, and I have updated the paper dataset,and may feel that the content is too much to be elaborated in the article. I still have many problems to solve in my research, and I would like to consult and learn from you.
  •  
  • (3) Regarding the dataset annotations, clarity is essential. My experience suggests that when working with object detection frameworks like YOLO, human annotation is crucial for generating ground truth. Were multiple annotators involved in this process? And if so, was there an established criterion or standard to ensure consistency in labeling the defects? Since YOLO operates on the principle of supervised learning, ensuring that data is Independent and Identically Distributed (i.i.d.) is of paramount importance. The absence of this detail in the manuscript raises questions about the reliability of the presented results. I strongly recommend the authors to address this critical aspect by adding more details about the labeling process to enhance the credibility of the study.
  •  
  • Answer: We have expanded the content of section 4.1, including adding a table and some pictures to illustrate the standards followed when creating the dataset. And it shows the actual number of boxes for each type in the dataset, all between 270 and 320, which will not cause any problem of disrupting category balance during training.
  • (4) Regarding the Literature review section, I observed that the content might benefit from more recent updates. While one-stage and two-stage object detection frameworks were indeed considered state-of-the-art in 2020, given that we're now in 2023, it's pivotal to consider recent advances in the domain. In particular, Transformer-based object detection and segmentation models have gained significant attention. For instance, the Segment Anything Model (SAM) from Meta AI presents a compelling approach to object detection. I would highly recommend the authors to update their literature review section to incorporate these recent developments. Additionally, considering a comparative evaluation with the Segment Anything Model (SAM) might elevate the novelty of your work, potentially capturing the interest of a broader readership.

  • Answer:   I have updated the literature review section to include these recent developments.
  • (5) In reference to the multiple-source space attention module depicted in Figure 5, I'm curious about the rationale behind the chosen architecture. Is there a specific motivation or empirical evidence suggesting its superiority for the task at hand? Given the advancements in Neural Architecture Search (NAS), it's conceivable that more optimal architectures might be attainable. It would be enlightening to see additional empirical evidence or discussions that underline the effectiveness of this particular structure in achieving the paper's objectives.

  • Answer: Your attention is very in-depth. We did conduct in-depth research on the multiple source space attention module, but due to negligence, we overlooked this part in our writing. Now we have rewritten it in section 4.6.2. By comparing the multiple source space attention module with the ablation experiment of SAM, it can be concluded that as the number of attention modules increases, MSAM is more stable compared to SAM, and there is no sudden decrease in accuracy, and the accuracy of MSAM is positively correlated with the number of usage. This indicates that MSAM's multi information source scheme has played a role in achieving coordinated consideration when adjusting spatial weights.
  • (6) In lines 285 to 287, the manuscript states, “Set the IOU threshold to 0.5, which means that a prediction box is considered a correct prediction only if the intersection ratio between the true box and the prediction box is greater than 0.5.” While this is a standard approach, I'd like to delve deeper into a scenario: Suppose there's a single ground truth box, and the model predicts four boxes. Assuming non-max suppression doesn't reduce them and each of these predicted boxes overlaps with the ground truth box with an IOU > 0.5, would they all be deemed correct? In conventional computer vision practices, this might be accepted since the emphasis is on prediction quality. However, in fields like electronics where physical interpretations are crucial, such outcomes might not be ideal. For instance, while one predicted box correctly matches the target, the other three could be perceived as noise, offering no additional informational value. It would be beneficial if the authors clarify their stance and specify whether they're adhering to the norms of the computer vision community in this regard.

  • Answer: As you mentioned, the problem of "multiple prediction boxes with IOU>0.5 are still retained after non maximum suppression of the prediction results" can be solved by adjusting the parameters of non maximum suppression and IOU threshold simultaneously. Because non maximum suppression is also judged and the results are given through IOU, when the IOU setting in non maximum suppression is less than the IOU threshold in the results, it can ensure that there is only one prediction box in the same position (in the same category). In addition, if the defects in the dataset are very dense, it will increase the probability of the occurrence of "one real box corresponding to multiple prediction boxes". However, the high-frequency transformer dataset used in this study has relatively sparse defects. From this set of data with "5416 images and 6972 real boxes", it can be seen that the average number of defects in each image is less than 2, so there will not be multiple effective prediction boxes during prediction. This also matches the actual production situation, as the current yield rate of automated production is very high, and most rejected products only have one defect, so there will not be a situation where a large number of defects are concentrated.
  • (7) In the manuscript, I did not identify any specific mention or citation regarding the YOLO v7 framework upon which your model appears to be constructed. I strongly encourage the authors to duly acknowledge the source of the code they adapted to develop their model. If the model was developed from scratch, it would be surprising. Training a model from scratch, especially on comprehensive datasets like COCO, is both time-consuming and resource-intensive. Proper citation and acknowledgment of prior work are crucial for scholarly integrity.

  • Answer: I added YOLOv7 references.

  • Below are some highlighted grammatical errors besides the title error mentioned above; however, this isn't an exhaustive list. I strongly recommend a thorough proofreading of the document.

    (1)In the abstract, lines 9 to 13, please avoid capitalizing words after a comma. Additionally, the sentences in this section require restructuring for clarity and coherence.

  • Answer:I have made the necessary modifications.
  •  

    (2)While it's a minor point, I noticed some authors have used email addresses from domains like 163.com and qq.com. Considering all authors are affiliated with Xi’an University of Architecture and Technology, is there a specific reason for not using educational email addresses?

  • Answer:I have made the necessary modifications. zqkang@xauat.edu.cn
  •  

    (3) In line 31, it appears that the Chinese character “、” has been used. Please replace it with the standard English comma for consistency.

  • Answer:I have made the necessary modifications.
  •  

    (4) In line 115, the sentence "Duong Huong Nguyen[30] The Faster R-CNN algorithm is used to detect structural defects of steel plates." should be revised for clarity. Perhaps it could read: "Duong Huong Nguyen [30] employed the Faster R-CNN algorithm to detect structural defects in steel plates."

  • Answer:I have made the necessary modifications.
  •  

    (5) In line 161, the phrase "Can retain the most effective features." should be revised for completeness. It might read: "The proposed model can retain the most effective features."

  • Answer:I have made the necessary modifications.
  •  

    (6) In lines 257 to 253, the mathematical notations "? 2 (?, ? ??)", "w^gt", and "h^gt" appear to be incomplete or incorrectly formatted. Please ensure that these notations are properly presented and defined in the text.

  • Answer:I have made the necessary modifications.

Reviewer 3 Report

The manuscript introduces a novel DME-YOLO structure designed for high-frequency transformer appearance defect detection. It addresses the trade-off between detection accuracy and speed in traditional YOLO models. The proposed DME-YOLO model is based on YOLOv7 and incorporates Dense blocks in the backbone to reduce parameters and floating point computations. Additionally, a multi-source attention mechanism module called MSAM is introduced to enhance spatial information integration, aiming to improve detection accuracy for small target defects. The manuscript reports promising test results, including an accuracy of 97.6 mAP and a speed of 51.2 FPS.

The manuscript addresses a relevant and significant problem - high-frequency transformer appearance defect detection. The trade-off between detection accuracy and speed is an important consideration in real-world applications. The proposed DME-YOLO structure, incorporating Dense blocks and MSAM, demonstrates innovation in improving both detection accuracy and speed. The manuscript presents promising experimental results, including a high accuracy of 97.6 mAP and a speed of 51.2 FPS. These results indicate the effectiveness of the proposed model.

Introduce more relevant studies on the application of CNN-based models for object detection, such as: doi.org/10.1007/s00170-022-10335-8 and doi.org/10.3390/s22207907.

Provide a more detailed and step-by-step explanation of how the proposed enhancements, such as Dense blocks, feature multiplexing, MSAM, and the EIOU loss function, are integrated into the YOLOv7 framework. This will help readers better understand their roles and significance.

Elaborate on the technical aspects of the introduced components, including how Dense blocks optimize the backbone, how MSAM integrates spatial information, and how the EIOU loss function affects target frame regression.

Consider including visual aids, such as diagrams or flowcharts, to illustrate the modifications and the flow of information within the proposed DME-YOLO structure.

In conclusion, the manuscript addresses a relevant and significant problem in high-frequency transformer appearance defect detection. The proposed DME-YOLO structure shows promise in improving both detection accuracy and speed. With the relevant comments to be addressed, the manuscript has the potential to make a meaningful contribution to the field of defect detection.

The quality of English language in the manuscript is generally good.

Author Response

(1)Provide a more detailed and step-by-step explanation of how the proposed enhancements, such as Dense blocks, feature multiplexing, MSAM, and the EIOU loss function, are integrated into the YOLOv7 framework. This will help readers better understand their roles and significance.

Answer:This paper provide a  detailed and step-by-step explanation of the proposed enhancements, such as Dense blocks, feature multiplexing, MSAM, and the EIOU loss function, are integrated into the YOLOv7 framework.

(2)Elaborate on the technical aspects of the introduced components, including how Dense blocks optimize the backbone, how MSAM integrates spatial information, and how the EIOU loss function affects target frame regression.

Answer:I have made the necessary modifications.

(3)Consider including visual aids, such as diagrams or flowcharts, to illustrate the modifications and the flow of information within the proposed DME-YOLO structure.

Answer:After the paper was revised, it was explained.

Reviewer 4 Report

A novel YOLO based approach is presented for object detection and the idea is clear. However,  there exists some issues to be addressed. 

1. In the introduction,  it will be better to introduce more deep learning and YOLO based approaches on various datasets.

2. Please indicate the limitations of proposed approach along with justification of its superiority. 

3. Including more visualization may be beneficial to illustrate superiority of the proposed approach.

4. Explain in detail about parameter tuning and the validation of the method.

English edit is required throughout the manuscript. 

Author Response

  1. In the introduction,  it will be better to introduce more deep learning and YOLO based approaches on various datasets.

        Answer:After the paper was revised, it was explained and more references                         were introduced.

  1. Please indicate the limitations of proposed approach along with justification of its superiority.

          Answer:When the amount of data is small, the superiority of this method             cannot be well reflected. The larger the amount of data, the better it can               be  reflected.

  1. Including more visualization may be beneficial to illustrate superiority of the proposed approach.

          Answer:This article has added some visualization diagrams and tables                  during the modification,it can beneficial to illustrate superiority of the                  proposed approach.

  1. Explain in detail about parameter tuning and the validation of the method.

         Answer: After revising this article, explain in detail about parameter tuning           and the validation of the method.

Round 2

Reviewer 2 Report

The revised manuscript looks fine now.

Good

Reviewer 4 Report

The authors have addressed all my concerns. It is ready for the publication. 

Minot English edit is required.